# Influence of Water and Fertilizer Reduction on Respiratory Metabolism in Sugar Beet Taproot (*Beta vulgaris* L.)

**DOI:** 10.3390/plants13162282

**Published:** 2024-08-16

**Authors:** Yuxin Chang, Guolong Li, Caiyuan Jian, Bowen Zhang, Yaqing Sun, Ningning Li, Shaoying Zhang

**Affiliations:** 1College of Agronomy, Inner Mongolia Agricultural University, Hohhot 010019, China; changyuxin0912@163.com (Y.C.); lgl9@sina.com (G.L.);; 2Special Crops Institute, Inner Mongolia Academy of Agricultural & Animal Husbandry Science, Hohhot 010019, China

**Keywords:** sugar beet, respiration rate, respiratory enzyme activity, gene expression, energy charge

## Abstract

Inner Mongolia, a major region in China for growing sugar beet, faces challenges caused by unscientific water and fertilizer management. This mismanagement restricts the improvement of sugar beet yield and quality and exacerbates water waste and environmental pollution. This study aims to evaluate the effects of reduced water and fertilizer on the growth and physiological metabolism of sugar beet taproot. Field experiments were conducted in Ulanqab, Inner Mongolia, in 2022 and 2023, using a split-plot design with three levels each of fertilization and irrigation. The study analyzed the effects of reduced water and fertilizer treatments on fresh taproot weight, respiration rate, energy metabolism, respiratory enzyme activity, and gene expression in sugar beet taproot. It was found that a 10% reduction in fertilizer significantly increased the beet taproot fresh weight. Further research revealed that during the rapid leaf growth phase and the taproot and sugar growth period, a 10% reduction in fertilizer upregulated *HK* and *IDH* gene expression and downregulated *G6PDH* gene expression in the beet taproot. This increased HK and IDH activities, decreased G6PDH activity, enhanced the activity of the EMP-TCA pathway, and inhibited the PPP. Taproot weight was positively correlated with the respiration rate, ATP content, EC, and ATPase, HK, and IDH activities, thereby increasing the taproot growth rate and taproot fresh weight, with an average increase of 4.0% over two years. These findings introduce a novel method for optimizing fertilizer use, particularly beneficial in water-scarce regions. Implementing this strategy could help farmers in western Inner Mongolia and similar areas improve crop yield and sustainability. This study offers new insights into resource-efficient agricultural practices, highlighting the importance of customized fertilization strategies tailored to local environmental conditions.

## 1. Introduction

Sugar beet (*Beta vulgaris* L.), an important sugar and economic crop in North China, plays an indispensable role in the domestic sugar industry and agricultural economy [1]. The planting area of sugar beet in Inner Mongolia accounts for over 50% of the total area of sugar beet agriculture in China, making it the most concentrated region for this crop in the country [2]. Achieving high-yielding and high-quality sugar beets depends on adequate irrigation and appropriate fertilization management [3,4]. In recent years, continuous improvements in sugar beet productivity have led to increased irrigation and fertilization, resulting in excessive consumption of water and fertilizer resources and gradual soil degradation. This issue is particularly acute in the water-scarce western region of Inner Mongolia, where excessive use of water and fertilizers has become a major factor limiting sustainable agricultural development [5,6].

To address this challenge, reducing irrigation and fertilization has become essential to improve water and fertilizer use efficiency, protect soil quality, and promote sustainable agricultural development. Numerous studies have shown that moderate reductions in water and fertilizer can effectively alleviate water resource pressures, improve soil environments, stimulate crop growth potential, enhance stress resistance, and ultimately ensure or even increase crop yield and quality [7,8]. Therefore, exploring optimized water and fertilizer management strategies for sugar beets, particularly in water-scarce areas, has become a focal point of current research.

Respiratory metabolism is crucial for plant growth and development. Through respiration, cells continuously break down substances [9], providing the necessary energy for various plant life activities [10]. It also participates in important biochemical processes, serving as a hub for energy and material metabolism in plants [11]. Multiple metabolic pathways are involved in respiratory metabolism, and they may vary under different growth stages and environmental conditions [12]. Glycolysis (EMP) and the tricarboxylic acid cycle (the TCA cycle) are common oxidation pathways for carbohydrates, fats, and proteins. They are the primary energy sources for crop life activities and act as key connections for interaction and conversion with other metabolic pathways, playing a significant role in regulating crop metabolic activities [13,14,15].

The pentose phosphate pathway (PPP) is another crucial pathway in crop respiratory metabolism [16,17]. This pathway is regulated by two rate-limiting enzymes: 6PGDH and G6PDH [18]. When the EMP-TCA pathway is blocked, the PPP can be activated to replace normal aerobic respiration, thereby maintaining crop growth, development, and environmental adaptation. Studies have shown that the PPP is closely associated with crop growth, development, and responses to various environmental stresses [19]. The respiration rate, respiratory enzyme activity, ATP content, and EC level in sugar beet taproot are closely related to the fresh weight of the taproot [20].

Nitrogen fertilizer is closely linked to plant respiration [21]. When nitrogen fertilizer is appropriately reduced, plants increase the allocation of carbon to their roots to obtain sufficient nitrogen for growth and development [22]. This promotes root growth and increases the root respiration rate [23]. Li et al. (1996)’s study demonstrated that applying an appropriate amount of nitrogen fertilizer can increase PFK (phosphofructokinase) and G6PDH (glucose-6-phosphate dehydrogenase) activities in walnut root, while excessive or insufficient application can inhibit root respiration [24]. Lu (2014) demonstrated that ICDH (isocitrate dehydrogenase) is crucial for maintaining high and stable rice yields through the construction of an ICDH co-suppressed family in japonica rice [25]. Jesús et al. (1998) found that IDH activity and expression levels increase during tomato fruit ripening, indicating that IDH may be involved in nitrogen transport [26]. Studies on potatoes have shown that IDH plays a role in nitrogen absorption and recycling [27].

In rice, three *IDH*-family genes (*OsIDHa*, *OsIDHc;1*, *OsIDHc;2*) are expressed in the root [28]. Research on the *ZmDof* gene in rice found that increased nitrogen use efficiency in transgenic rice is related to an enhanced expression of *ICDH* genes [29]. Different phosphorus application rates can have varying impacts on respiratory metabolic pathways [30].

This region currently faces severe water shortages and resource waste issues, which are exacerbated by unscientific water and fertilizer management practices among farmers. To maximize efficiency, exploring the feasibility of reducing water and fertilizer usage is crucial. Respiration plays a critical role in determining crop yield. The impact of reduced water and fertilizer on the respiratory metabolism of sugar beet taproot remains unclear. Therefore, we aim to investigate the effects of reducing water and fertilizer on the respiratory metabolism of sugar beet. This study aims to reveal how water and fertilizer management strategies influence the energy metabolism of sugar beet crops, thereby affecting their growth and development, and to provide a theoretical basis for optimizing water and fertilizer use.

## 2. Results

### 2.1. Sugar Beet Taproot Growth

Based on the measurements of sugar beet fresh taproot weight in 2022 and 2023 (Figure 1), we observed differences in taproot weight among different treatments at various growth stages. Specifically, under reduced-water conditions, a 15% reduction in irrigation (F3W2) did not show a significant difference compared to the control, particularly at 80 and 120 days after transplanting. However, a 30% reduction in irrigation (F3W1) significantly reduced the taproot weight compared to the control (CK). Under reduced-fertilizer conditions, at 40, 80, and 120 days after transplanting, we found that a 10% reduction in fertilization (F2W3) had increased the taproot weight compared to CK, with average increases of 3.4%, 4.7%, and 4.0% over the two years. Notably, in 2023, the effect of reduced fertilizer was more pronounced, resulting in a significant increase in taproot weight. However, when fertilization was reduced by 20%, all treatments showed a significant decrease in taproot weight at various time points. At 120 days after transplanting, when both water and fertilizer supplies were reduced, all treatments except the one with a 15% reduction in irrigation and the one with a 10% reduction in fertilization (F2W2) showed a significant decrease in taproot weight compared to CK. These results indicate that changes in sugar beet taproot weight are significantly affected by water and fertilizer management strategies. A moderate reduction in fertilizer supply can promote an increase in taproot weight, and a moderate reduction in water supply does not significantly affect taproot weight/growth. However, excessive reductions in both the water supply and the fertilizer supply lead to a significant decrease in taproot weight.

### 2.2. Respiration Rate of Sugar Beet Taproot

The respiration rate of the sugar beet taproot initially increased and then decreased throughout the growth cycle, peaking at 80 days after transplanting (Figure 2). This trend was consistent over the two-year observation period. Specifically, under reduced-water-supply conditions, data at 40 and 80 days after transplanting showed that a 15% reduction in irrigation (F3W2) had not significantly affected the respiration rate compared to the control (CK), while a 30% reduction in irrigation (F3W1) had significantly decreased the respiration rate with an average reduction of 10% and 6% over the two years. Under reduced-fertilizer conditions, compared to CK, we found at 30 days after transplanting that a 10–20% reduction in fertilization (F2 and F1) had significantly decreased the respiration rate by 16–22%. At 40 and 80 days after transplanting, a 10% reduction in fertilization (F2W3) had increased the respiration rate, but a 20% reduction (F1W3) had significantly decreased it, particularly in 2023. At 120 days after transplanting, the respiration rate in the F2W3 treatment had significantly decreased by 7.4%. When both water and fertilizer supplies were reduced, the respiration rate in the F2W2 treatment did not differ significantly from CK at any growth stage. However, other reduced-water-and-fertilizer treatments showed varying degrees of decreased respiration rates at 40 and 80 days after transplanting. Overall, these results reveal that water and fertilizer management strategies significantly affect the respiration rate of sugar beet taproot, especially in the mid-to-late growth stages. A moderate reduction in fertilizer supply can increase the respiration rate at certain stages, but excessive reductions in water and fertilizer supplies lead to a decrease in the respiration rate.

### 2.3. Energy Metabolism of Sugar Beet Taproot

#### 2.3.1. Contents of ATP, ADP, and AMP

This study investigated the effects of water and fertilizer management on the ADP, ATP, and AMP contents in sugar beet taproot (Figure 3). The results showed that the ATP content in the sugar beet taproot first increased and then decreased during the growth cycle, peaking at 80 days after transplanting. Different water and fertilizer treatment strategies significantly affected the contents of these energy molecules at various growth stages. Under reduced-water conditions (Figure 3A), the ATP content in the F3W2 treatment did not show significant differences at 40, 80, and 120 days after transplanting compared to the control (CK), while the ATP content in the F3W1 treatment had significantly decreased by 5.8–12.9%. Under reduced-fertilizer conditions, at 30 days after transplanting, ATP content in the F2 and F1 treatments had significantly decreased by 9.1–11.7% compared to CK. At 40 and 80 days after transplanting, the F2W3 treatment showed a significantly increased ATP content, while that of the F1W3 treatment had significantly decreased. At 120 days after transplanting, all treatments showed a significant decrease in ATP content by 11.8% and 6.8%. When both the water and fertilizer supplies were reduced, the ATP content in the F2W2 treatment did not significantly differ from that in the CK crops in the later growth stages, while other treatments showed a significant decrease.

Compared to the ATP results, changes in the ADP content followed a different trend (Figure 3B), initially decreasing and then increasing, reaching the lowest level at 40 days after transplanting. Under reduced-water conditions, at 80 and 120 days after transplanting, the ADP content in the F3W1 treatment had significantly increased by 6.7% and 7.4%, respectively, compared to CK. Regarding the reduced-fertilizer treatments, the ADP content in the F1W3 treatment significantly increased by 9.4–11.2% throughout the growth cycle, while ADP content in the F2W3 treatment was found to be decreased at 40 and 80 days after transplanting but significantly increased at 120 days. When both the water and fertilizer supplies were reduced, ADP content in the F2W1, F1W2, and F1W1 treatments was significantly increased compared to CK in the later growth stages.

Regarding AMP (Figure 3C), under reduced-water conditions, the AMP content in the F3W2 treatment was lower than the CK level throughout the growth cycle, while the AMP content in the F3W1 treatment was higher than the CK level. Under reduced-fertilizer conditions, AMP content in the F2W3 treatment was found to be decreased at 40 and 80 days after transplanting but significantly increased at 120 days. AMP content in the F1W3 treatment was significantly increased throughout the growth cycle. When both water and fertilizer supplies were reduced, AMP content in the F2W1, F1W2, and F1W1 treatments increased compared to CK at all measured times.

#### 2.3.2. Energy Charge

Energy charge (EC) is a direct indicator of the cellular energy status. We found that the EC of our sugar beet taproot initially increased and then decreased during the growth cycle, peaking at 40 and 80 days after transplanting, indicating the highest energy status within the cells at these stages (Figure 3D). Under reduced-water-supply conditions, the EC in the F3W2 treatment did not significantly differ from that of CK at 40, 80, and 120 days after transplanting, indicating that a moderate reduction in water supply had limited impact on the energy status of the sugar beet cells. However, the EC in the F3W1 treatment was significantly decreased by 2–5.4% compared to CK at these growth stages, highlighting the negative impact of excessive water reduction on the energy status of sugar beet cells. Under reduced-fertilizer-supply conditions, the EC in the F2W3 treatment had significantly increased by 4.3% and 1.9% at 40 and 80 days after transplanting, indicating that a moderate reduction in fertilizer supply can enhance the energy status within the cells during these periods. Conversely, the EC in the F2W3 treatment had significantly decreased by 3.3% and 9% at 30 and 120 days after transplanting. Additionally, the EC in the F1W3 treatment was significantly lower than that of CK at all growth stages, reflecting the inhibitory effect of substantial fertilizer reduction on the energy levels of sugar beet cells. When both the water and fertilizer supplies were reduced, the EC in the F2W2 treatment did not significantly differ from that of CK at any growth stage, indicating that sugar beets can adapt to mild reductions in water and fertilizer to some extent. However, the EC in the F2W1, F1W2, and F1W1 treatments was significantly lower than that of CK at all measured growth stages, underscoring the significant negative impact of stricter water and fertilizer restrictions on the energy status of sugar beet cells.

#### 2.3.3. ATPase

The reductions in water and fertilizer significantly affected the ATPase activity in the sugar beet taproot (Figure 3E). Compared to the control, ATPase activity increased in the F3W2 treatment under reduced water supply, but decreased in the F3W1 treatment. This suggests that a moderate reduction in water supply may activate energy metabolism mechanisms in sugar beets, enhancing ATPase activity, while more severe water restrictions inhibit enzyme activity. Under reduced-fertilizer conditions, the ATPase activity in the F2W3 treatment was significantly higher than the CK level at 40 and 80 days after transplanting, indicating that moderate fertilizer reduction helps improve energy metabolism efficiency in sugar beets. However, the ATPase activity in the F1W3 treatment was significantly lower than the CK level at 40 and 120 days after transplanting, suggesting that excessive fertilizer reduction may negatively affect energy metabolism in sugar beets. When both the water and fertilizer supplies were reduced, ATPase activity in the F2W2 treatment was increased at 40 and 80 days after transplanting compared to CK, while ATPase activity in the F2W1, F1W2, and F1W1 treatments was decreased at all growth stages. Simultaneous reductions in the water and fertilizer supply significantly inhibited the energy metabolism of the sugar beets.

### 2.4. Key Enzymes in Respiratory Metabolism of Sugar Beet Taproot

#### 2.4.1. Hexokinase

As a key enzyme in the glycolysis pathway, changes in hexokinase (HK) activity are crucial for understanding plant energy and respiratory metabolism in response to environmental regulation. Our results reveal dynamic changes in HK activity under different water and fertilizer treatments (Figure 4A). Under reduced water supply, HK activity in the F3W2 treatment was increased compared to the control at 40 and 80 days after transplanting, while it was decreased in the F3W1 treatment. However, at 120 days after transplanting, this trend reversed, and HK activity in the F3W1 treatment increased compared to CK.

Under reduced fertilizer supply, HK activity in the F2W3 treatment was significantly increased by 6.3% at 80 days after transplanting compared to CK, while it was significantly decreased in the F1W3 treatment. At 120 days after transplanting, HK activity in the F2W3 treatment was significantly decreased, while it was significantly increased in the F1W3 treatment. When both the water and fertilizer supplies were reduced, HK activity in the F2W2 treatment did not significantly differ from CK at any growth stage, indicating that mild reductions in water and fertilizer have limited impact on HK activity. However, HK activity in the F2W1, F1W2, and F1W1 treatments was decreased by 2.3–9.7 at 40 and 80 days after transplanting but significantly increased by 5.9–8.2% at 120 days. This suggests that simultaneous reductions in water and fertilizer may inhibit HK activity in the early and mid-growth stages but activate it in the late growth stage.

The gene expression levels of HK in taproot (Figure 5A) showed trends consistent with enzyme activity, indicating that HK responds to water and fertilizer changes at the transcriptional level.

#### 2.4.2. Isocitrate Dehydrogenase

Isocitrate dehydrogenase (IDH), a key metabolic enzyme in the TCA cycle, plays a crucial role in energy production and respiratory metabolism. Data analysis revealed the following (Figure 4B): Compared to CK, under reduced-water-supply conditions, IDH activity in the F3W2 treatment did not significantly change throughout the growth cycle, indicating that sugar beets in this treatment could better maintain IDH activity stability. In the F3W1 treatment, IDH activity was significantly decreased by 1.7% at 80 days after transplanting, while it was increased at 120 days.

Under reduced fertilizer supply, IDH activity in the F2W3 treatment was significantly increased by 4.7% at 80 days after transplanting, while it was significantly decreased by 4.9% in the F1W3 treatment. This suggests that a moderate reduction in fertilizer supply can promote IDH activity, benefiting sugar beet energy metabolism. In contrast, excessive reduction inhibits IDH activity. At 120 days after transplanting, this trend was reversed, revealing the complexity of the long-term effects of different fertilizer management strategies on IDH activity. This indicates that excessively reducing the fertilizer supply inhibits IDH activity in the short term but may promote compensatory physiological adjustments in the long term.

When both water and fertilizer supplies were reduced, IDH activity in the F2W2 treatment did not significantly differ from the CK level at any growth stage, indicating that mild reductions in water and fertilizer do not significantly affect IDH activity. IDH activity in the F2W1, F1W2, and F1W1 treatments was significantly decreased by 3.5–5.8% at 80 days after transplanting but increased at 120 days, suggesting that simultaneous reductions in water and fertilizer inhibit IDH activity in the mid-growth stages but may lead to metabolic adaptation in later stages.

Measurements of the IDH gene expression levels (Figure 5B) showed trends consistent with enzyme activity at various growth stages, indicating that IDH responds to water and fertilizer changes at the transcriptional level.

#### 2.4.3. Glucose-6-Phosphate Dehydrogenase

Glucose-6-phosphate dehydrogenase (G6PDH) plays a key role in the pentose phosphate pathway and is crucial for cellular antioxidant defense and providing reducing power. Comparing G6PDH activity changes under different water and fertilizer treatments (Figure 4C), we found that, compared to the control, G6PDH activity had increased in the F3W1 treatment at 40, 80, and 120 days after transplanting, suggesting that enhanced G6PDH activity helps cope with water stress. Under reduced-fertilizer conditions, G6PDH activity in the F2W3 treatment decreased compared to CK, while it significantly increased in the F1W3 treatment at all growth stages. When both water and fertilizer supplies were reduced, G6PDH activity in the F2W2 treatment did not significantly differ from CK at any growth stage, while it increased in the F2W1, F1W2, and F1W1 treatments, with F1W1 showing more significantly increased activity at 40 and 80 days after transplanting. This indicates that excessive reductions in water and fertilizer might activate the pentose phosphate respiratory metabolism pathway in sugar beets.

Measurements of G6PDH gene expression levels (Figure 5C) showed trends consistent with enzyme activity at various growth stages, indicating that G6PDH responds to water and fertilizer changes at the transcriptional level.

### 2.5. Correlation Analysis between Taproot Fresh Weight and Respiratory Metabolism Indicators

An in-depth analysis was conducted on the correlation between sugar beet taproot fresh weight and energy substances and respiratory enzymes at different growth stages (Figure 6). At 40 and 80 days after transplanting, the sugar beet taproot fresh weight showed a highly significant positive correlation with the ATP, EC, ATPase, and isocitrate dehydrogenase (IDH) data. This indicates that during the rapid leaf growth stage and the taproot and sugar growth stage of sugar beets, the increase in taproot fresh weight is closely related to improved cellular energy status, enhanced energy conversion efficiency, and increased TCA cycle activity. Additionally, the taproot fresh weight showed a positive correlation with the respiration rate and hexokinase (HK) activity, reflecting the positive relationship between taproot growth and respiration. Meanwhile, the taproot fresh weight showed a highly significant negative correlation with ADP, AMP, and glucose-6-phosphate dehydrogenase (G6PDH). During the vigorous growth period of sugar beets, a higher taproot fresh weight is associated with lower ADP and AMP levels (i.e., higher energy efficiency) and a relative decrease in the pentose phosphate pathway’s functioning. At 120 days after transplanting, the sugar beet taproot fresh weight showed a highly significant negative correlation with HK, IDH, and G6PDH. This indicates that during the sugar accumulation period, a higher taproot fresh weight is associated with reduced activity of the EMP, the TCA cycle, and the pentose phosphate pathway, reflecting that weakened taproot respiratory metabolism in the later growth stage is beneficial for taproot growth.

## 3. Discussion

This study demonstrates that a 10% reduction in fertilization and a 15% reduction in irrigation can be applied to enhance crop production. Specifically, a 10% reduction in fertilization (F2W3) significantly increases sugar beet yield, indicating potential for further reductions in water and fertilizer use. However, excessive reductions in water and fertilizer supply decrease the taproot fresh weight, particularly in the later growth stages. This may be due to nutrient and water stress caused by the reduced water and fertilizer, which affects the normal physiological activities and growth of sugar beets. Studies by Chen et al. (2023) and Wei et al. (2021) show that appropriate water and fertilizer supply increases the yield of apples and corn, while excessive or insufficient supply reduces yield, consistent with the results of this study [31,32]. However, the study by Guo Ming et al. (2024) indicates that under low nitrogen, taproot weight increases during the taproot and sugar growth period [33]. In contrast, this study shows that this increase starts during the rapid leaf growth period. This difference is mainly due to fertilizer type; in this study, nitrogen, phosphorus, and potassium fertilizers were simultaneously reduced, regulating taproot growth and development sooner compared to reducing only nitrogen fertilizer.

The results show that the respiration rate of the sugar beet taproot initially increased and then decreased during the growth cycle, significantly affected by water and fertilizer. The F2W3 treatment could increase the respiration rate in the mid-growth stage, likely because moderate fertilizer reduction improves nutrient use efficiency, enhancing energy metabolism. However, insufficient water and fertilizer supply may result in inadequate energy production, affecting plant growth and development. This is consistent with studies by Vogel et al. (2005) and Li et al. (2019), indicating that suitable environmental conditions are crucial for maintaining the high respiration rates necessary for crop growth [23,24]. Respiration is the foundation of all life activities, providing energy for taproot growth [34]. Higher respiration rates enable crops to produce more energy, maintaining growth [35]. In this study, we found that the F2W3 treatment had significantly increased the respiration rate, ATP content, EC, and ATPase activity at 40 and 80 days after transplanting, with increased activity at 120 days. When water and fertilizer were excessively reduced, the trends reversed at all stages. This is similar to the findings of Yu et al. (2023). in sugar beets [36]. Our study also found significant correlations between ATP, ADP, and AMP contents, EC in sugar beet taproot, and taproot fresh weight, revealing the direct impact of energy metabolism on sugar beet growth and development. The trend of the ATP content first increasing and then decreasing, along with its positive correlation with taproot fresh weight, emphasizes ATP’s important role as an energy source in supporting taproot growth. Additionally, changes in the EC reflect cellular energy status, and its association with taproot fresh weight highlights the importance of maintaining a high energy state for promoting sugar beet taproot development. This result is similar to the findings of Xu et al. (2019) in rice [37].

In this study, a 10% reduction in fertilization increased glycolysis by promoting HK activity and enhanced TCA cycle efficiency by increasing IDH activity, thereby boosting energy production to support sugar beet growth. However, excessive reductions in the water and fertilizer supplies decreased HK and IDH activities, potentially slowing energy production and limiting growth and development. The changes in activity were consistent with changes in HK and IDH gene expression levels, similar to findings by Vogel et al. (2005) in rice [23]. This further confirms the impact of water and fertilizer management on enzyme activity regulation and emphasizes the importance of adjusting respiratory metabolism pathways in response to water and fertilizer changes.

Increased G6PDH activity may enhance the PPP, providing more reducing power and energy for sugar beets, especially under insufficient water and fertilizer supply, thereby improving the plant’s ability to adapt to environmental stress. The regulation of G6PDH gene expression highlights the importance of water and fertilizer management in activating plant stress responses and adjusting energy metabolism. This study found that with moderate fertilizer reduction, EMP-TCA pathway enzyme activity increased while PPP enzyme activity decreased. Conversely, with excessively reduced amounts of water and fertilizer, EMP-TCA pathway enzyme activity decreased while PPP enzyme activity increased. This indicates that different water and fertilizer management strategies alter respiratory pathways, consistent with the findings of Luo et al. (2007) [38].

The results of this study underscore the significant impact of water and fertilizer management on sugar beet taproot fresh weight, respiration rate, energy metabolism, and respiratory metabolism. These findings provide empirical evidence for optimizing sugar beet growth by adjusting water and fertilizer supply. Future research can further explore the interaction mechanisms between water and fertilizer management and the physiological metabolism of sugar beets. Additionally, it can examine how advanced agricultural technologies can achieve more precise water and fertilizer management to enhance crop yield and sustainability.

## 4. Materials and Methods

### 4.1. Experimental Site Description

This study was conducted from 2022 to 2023 at the Academy of Agriculture and Forestry Sciences in Ulanqab, Inner Mongolia, located at 112°27′ E, 40°26′ N. This region has a temperate arid continental monsoon climate, characterized by low rainfall and large seasonal temperature variations. During the study period, the average maximum temperatures from May to October were 26.9 °C in 2022 and 27.3 °C in 2023, while the average minimum temperatures were 4.3 °C and 6.4 °C, respectively. The total precipitation for these months was 360 mm in 2022 and 376 mm in 2023. The soil type is Calcic Chernozem and the soil texture is sandy loam. Soil nutrient conditions at the experimental site are detailed in Table 1.

### 4.2. Experimental Design

This study was conducted under fertigation conditions using a split-plot design, using the locally recommended fertilization and irrigation levels as the control group treatment (CK) (Table 2). The experiment was divided into main plots for fertilization levels and subplots for irrigation levels. Two reduced-fertilization treatments were set: 10% reduction (F2) and 20% reduction (F1). Irrigation treatments were set at 15% reduction (W2) and 30% reduction (W1). Each plot measured 6 m by 5 m, and the entire experiment was repeated three times. Seedlings were raised in paper tubes and transplanted, with an inter-row spacing of 50 cm and an inter-plant spacing of 18 cm. Nitrogen fertilizer application was divided into base and top dressing, with the base application accounting for 30% of the total nitrogen; top dressing was applied in three stages, with proportions of 30%, 30%, and 10%. Of the phosphorus and potassium fertilizers, a 50% proportion was applied as the base fertilizer, and top dressing was applied in three stages, with proportions of 20%, 20%, and 10%. The selected material was the locally grown variety IM1162. The fertilizers used were urea (N 46%), diammonium phosphate (N 18%, P_2_O_5_ 46%)/ammonium polyphosphate (N 17%, P_2_O_5_ 58%), and potassium sulfate (K_2_O 50%).

A Venturi system was used for fertigation to ensure uniform fertilizer application. The first irrigation was conducted immediately after sugar beet transplantation, followed by irrigation and fertigation 30, 70, and 110 days post transplantation. The irrigation amount for each treatment was precisely controlled using water meters (DN50, Beijing Shunshui Water Meter Factory, Beijing, China).

Detailed records of irrigation and fertilization amounts for each treatment were maintained, with specific data presented in Table 2. Except for specific water and fertilizer management measures, other field practices followed traditional sugar beet cultivation methods in the region.

### 4.3. Collection of Plant Samples

Samples were collected four times during the entire growth period of the sugar beets. Sampling times were 30 days after transplanting (seedling stage), 40 days after transplanting (rapid leaf growth stage), 80 days after transplanting (taproot and sugar growth stage), and 120 days after transplanting (sugar accumulation stage). Each treatment was repeated three times, and for each repetition, three uniformly growing sugar beet plants were selected. From each plant, a 5 g fresh taproot sample was taken, and the samples from the three plants were mixed thoroughly. Collected taproot samples were immediately frozen in liquid nitrogen and then stored in a −80 °C freezer.

### 4.4. Respiration Rate

The respiration rate was measured using an infrared gas analyzer (YX-306B, Beijing Yuxiang Hengye Measurement and Control Technology Co., Ltd., Beijing, China). The calculation formula used is as follows:(1)respiration rate=C/(W×t),
where *C* is the amount of CO_2_ (mg) released during measurement minus the initial CO_2_ release amount (mg), *W* is the fresh weight of the sugar beet taproot (g), and *t* is the measurement time (h) [40].

### 4.5. Respiratory Metabolic Enzyme Activity and Energy Measurement

Enzyme-linked immunosorbent assay (ELISA) kits were used to measure the activities of ATPase, PK, HK, IDH, SDH, 6PGDH, and G6PDH, as well as the contents of ATP, ADP, and AMP (Jiangsu Su Enzyme Science and Technology Co., Ltd., Taizhou, Jiangsu, China). The measurement of these parameters was conducted according to the kit instructions, the specific method used is as follows:

Under liquid-nitrogen-freezing conditions, grind the plant tissue and mix 0.1 g of fresh sample with 900 μL of 0.1% PBS buffer. After thorough shaking, centrifuge the tissue at 12,000 rpm for 30 min in a low-temperature refrigerated centrifuge. Collect the supernatant for enzyme activity assays.

(1)Standard Addition: Set up wells for the standards and samples. Add 50 μL of standards with different concentrations to each standard well.(2)Sample Addition: Set up blank wells (i.e., wells without the samples and enzyme reagents, but with all other steps remaining the same) and wells for the test samples. Add 40 μL of sample diluent to each test sample well on the coated microplate, followed by 10 μL of the test sample (resulting in a final dilution of 5 times). Gently mix.(3)Enzyme Addition: Add 100 μL of enzyme conjugate to each well except the blank wells.(4)Incubation: Cover the plate with a sealing film and incubate at 37 °C for 60 min.(5)Washing: Carefully remove the sealing film and discard the liquid. Fill each well with washing solution, let it stand for 30 s, and then discard the washing solution. Repeat this step five times and pat dry.(6)Color Development: Add 50 μL of chromogen solution A to each well, followed by 50 μL of chromogen solution B. Gently shake to mix, and incubate at 37 °C in the dark for 15 min.(7)Termination: Add 50 μL of stop solution to each well to terminate the reaction (this will cause the color to change from blue to yellow).(8)Measurement: Set the blank well to zero, and measure the absorbance (OD value) of each well at 450 nm. This should be completed within 15 min after adding the stop solution.(9)Calculation: Using the concentration and OD values of the standards, create a standard curve with a linear regression equation. Insert the OD values of the samples into this equation to calculate their concentrations, then multiply by the dilution factor to obtain the actual concentration of the samples.

### 4.6. Energy Charge Calculation Formula

The energy charge was calculated as follows:(2)$$EC=\frac{ATP+\frac{1}{2}ADP}{ATP+ADP+AMP}\;.$$

### 4.7. Total RNA Extraction and cDNA Synthesis from Sugar Beets

Total RNA was extracted using an RNA Kit (Jiangsu Kangwei Century Biological Technology Co., Ltd., Taizhou, China) according to the manufacturer’s instructions. RNA with 2.15 ≥ A260/A280 ≥ 1.95 and A260/A230 ≥ 2.0 was selected for reverse transcription to obtain cDNA (Jiangsu Kangwei Century Biological Technology Co., Ltd., Taizhou, China), which was diluted five times for use. Detailed steps followed the method of Guo et al. [41].

### 4.8. Real-Time Quantitative PCR

cDNA from sugar beet taproot at four stages was used as templates, with Actin as the internal reference gene, and PerfectStart Green qPCR SuperMix dye (TransGen Biotech Co., Ltd., Guangzhou, China) was added. The amplification program was as follows: 95 °C for 2 min, 95 °C for 10 s, 56 °C for 10 s, and 72 °C for 1 min, for 40 cycles. Each stage included three biological replicates and three technical replicates. Relative gene expression was calculated using the 2^−ΔΔCt^ method [42]. Primer sequences are shown in Table 3.

### 4.9. Statistical Analysis

All data are presented as the mean ± standard deviation (SD). One-way analyses of variance (ANOVAs) and Duncan’s multiple range test were conducted using SPSS version 20, with a significance level of *p* < 0.05. Each experiment included three replicates. Statistical analysis was conducted using Excel 2019, and plotting was carried out using Origin 2021 software.

## 5. Conclusions

Reducing fertilization by 10% (F2W3) enhanced metabolic processes in the beet taproot, leading to a 4.0% increase in taproot fresh weight over two years. This treatment specifically upregulated HK and IDH gene expression while downregulating G6PDH gene expression. Consequently, HK and IDH activities increased, G6PDH activity decreased, EMP-TCA pathway activity was enhanced, and the PPP was inhibited. These metabolic adjustments increased the taproot respiration rate, ATPase activity, ATP content, and energy charge (EC), all positively correlated with taproot fresh weight. These results indicate that a 10% reduction in fertilization supports sustainable agriculture by reducing fertilizer use while maintaining high yield and quality. Additionally, this fertilization strategy significantly improves the physiological efficiency of beet taproot. These findings can be applied in agriculture to optimize fertilization strategies, increase crop yield, and provide economic and environmental benefits.

## Figures and Tables

**Figure 1 plants-13-02282-f001:**
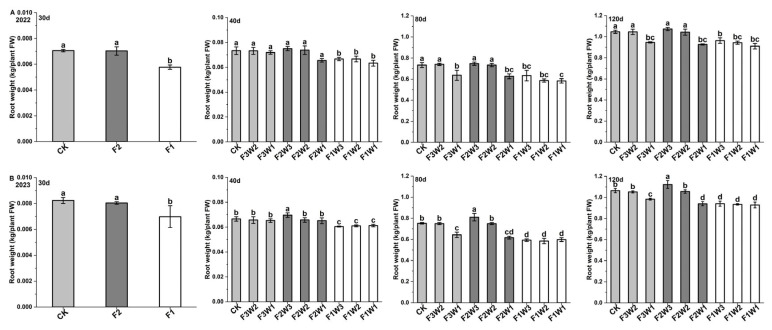
Impact of water and fertilizer reduction on fresh weight of sugar beet taproot: taproot respiration rates in (**A**) 2022 and (**B**) 2023; 30d: 30 days after transplanting—CK represents conventional basal fertilizer, F2 represents a 10% reduction in basal fertilizer, and F1 represents a 20% reduction in basal fertilizer; 40d: 40 days after transplanting, 80d: 80 days after transplanting, and 120d: 120 days after transplanting—CK (conventional water and fertilizer), F3W2 (irrigation reduced by 15%), F3W1 (irrigation reduced by 30%), F2W3 (fertilizer reduced by 10%), F2W2 (irrigation reduced by 15% and fertilizer reduced by 10%), F2W1 (irrigation reduced by 30% and fertilizer reduced by 10%), F1W3 (fertilizer reduced by 20%), F1W2 (irrigation reduced by 15% and fertilizer reduced by 20%), and F1W1 (irrigation reduced by 30% and fertilizer reduced by 20%). Letters indicate significance at the *p* < 0.05 level.

**Figure 2 plants-13-02282-f002:**
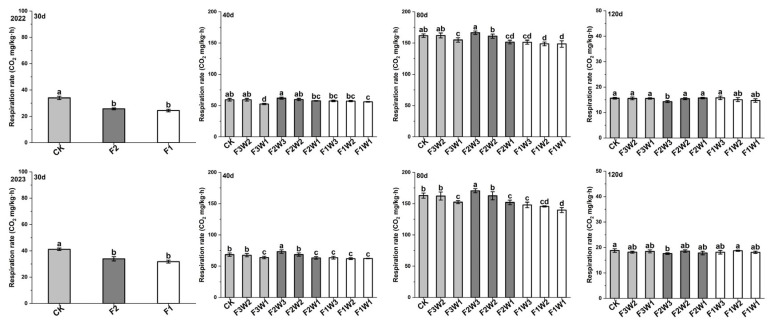
Impact of water and fertilizer reduction on respiration rate in sugar beet taproot, showing the respiration rates for 2022 and 2023; 30d: 30 days after transplanting, 40d: 40 days after transplanting, 80d: 40 days after transplanting, 120d: 40 days after transplanting. Letters indicate significance at the *p* < 0.05 level.

**Figure 3 plants-13-02282-f003:**
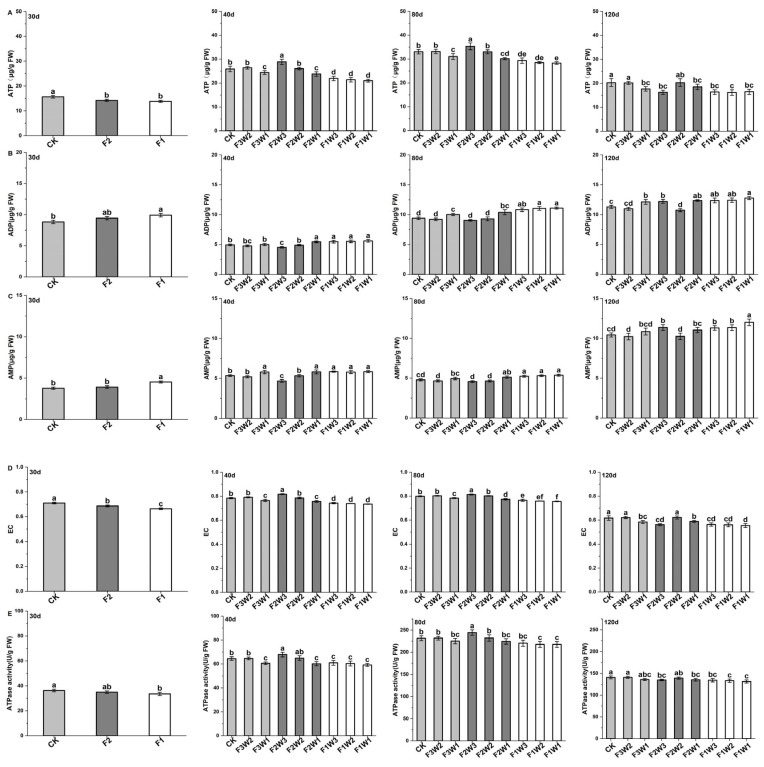
Impact of water and fertilizer reduction on energy metabolism in sugar beet taproot: (**A**) adenosine triphosphate (ATP), (**B**) adenosine diphosphate (ADP), (**C**) adenosine monophosphate (AMP), (**D**) energy charge, and (**E**) ATPase; 30d: 30 days after transplanting, 40d: 40 days after transplanting, 80d: 40 days after transplanting, 120d: 40 days after transplanting. Letters indicate significance at the *p* < 0.05 level. The experiment was conducted in 2022. The same applies to the following results.

**Figure 4 plants-13-02282-f004:**
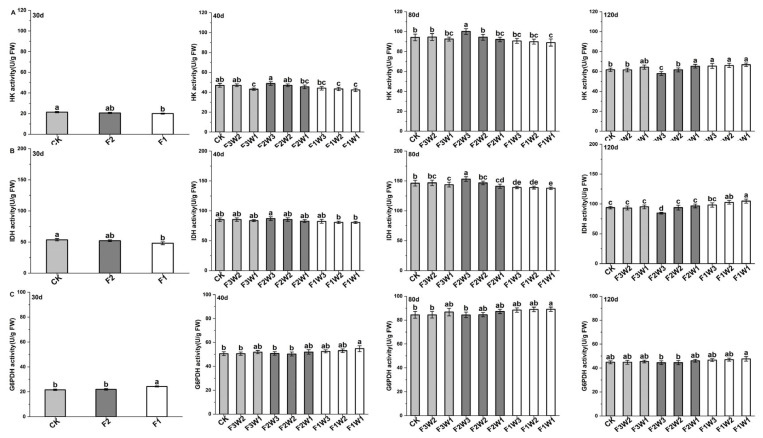
Impact of water and fertilizer reduction on the activity of key enzymes involved in respiratory metabolism in sugar beet taproot: (**A**) hexokinase, (**B**) isocitrate dehydrogenase, and (**C**) glucose-6-phosphate dehydrogenase; 30d: 30 days after transplanting, 40d: 40 days after transplanting, 80d: 40 days after transplanting, 120d: 40 days after transplanting. Letters indicate significance at the *p* < 0.05 level.

**Figure 5 plants-13-02282-f005:**
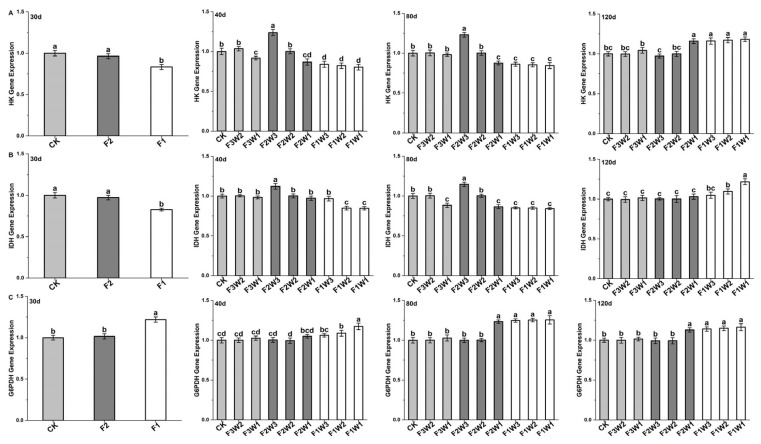
Impact of water and fertilizer reduction on the expression of respiratory metabolism enzyme genes in sugar beet taproot: (**A**) HK enzyme genes, (**B**) IDH enzyme genes, and (**C**) G6PDH enzyme genes; 30d: 30 days after transplanting, 40d: 40 days after transplanting, 80d: 40 days after transplanting, 120d: 40 days after transplanting. Letters indicate significance at the *p* < 0.05 level.

**Figure 6 plants-13-02282-f006:**
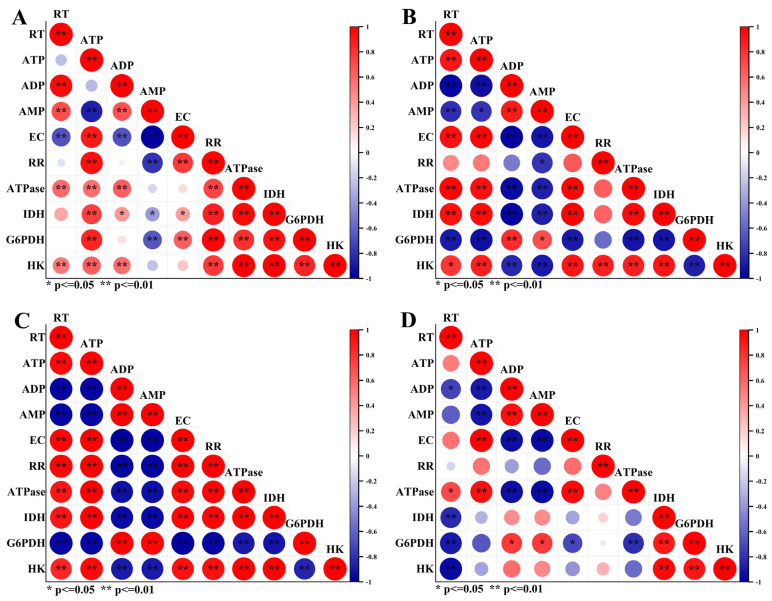
Correlation analysis between taproot fresh weight and respiratory metabolism indicators. RT: taproot weight, ATP: adenosine triphosphate, ADP: adenosine diphosphate, AMP: adenosine monophosphate, EC: energy charge, RR: respiration rate, ATPase: ATPase, IDH: isocitrate dehydrogenase, G6PDH: glucose-6-phosphate dehydrogenase, HK: hexokinase. (**A**) Whole growth period, (**B**) 40 days after transplanting, (**C**) 80 days after transplanting, and (**D**) 120 days after transplanting.

**Table 1 plants-13-02282-t001:** Soil condition of the test site.

Year	Total N (g·kg^−1^)	Total P (g·kg^−1^)	Total K (g·kg^−1^)	Available N (mg·kg^−1^)	Available P (mg·kg^−1^)	Available K (mg·kg^−1^)	pH Value	Organic Matter (g·kg^−1^)
2022	2.07	0.97	22.11	134.91	10.31	197.5	7.98	32.02
2023	1.71	0.46	16.31	111.07	9.23	153.01	7.71	18.21

According to Bo [39], the following methods were used for soil nutrient determination: total nitrogen (N)—determined using the Kjeldahl method; total phosphorus (P)—determined using the molybdenum blue colorimetric method; total potassium (K)—fused with NaOH (sodium hydroxide) and determined using flame photometry; available nitrogen—determined using the alkali diffusion method; available phosphorus—determined using the Olsen method; available potassium—extracted with NH_4_OAC (ammonium acetate) and determined using flame photometry.

**Table 2 plants-13-02282-t002:** The levels of fertilizer applied and the amount of irrigation administered in each treatment.

Treatment	Amount of Fertilizer Applied (kg/ha)			Total Irrigation Volume (m^3^/ha)
	Pure N	Pure P	Pure K	
F3W3(CK)	120	150	150	1350
F3W2				1147.5
F3W1				945
F2W3	108	135	135	1350
F2W2				1147.5
F2W1				945
F1W3	96	120	120	1350
F1W2				1147.5
F1W1				945

F3W3 (CK), F3W2, and F3W1—N: 120 kg/ha, P: 150 kg/ha, K: 150 kg/ha; F2W3, F2W2, and F2W1—N: 108 kg/ha, P: 135 kg/ha, K: 135 kg/ha; F1W3, F1W2, and F1W1—N: 96 kg/ha, P: 120 kg/ha, K: 120 kg/ha. Basal fertilization was applied using diammonium phosphate, top-dressing fertilization was applied using ammonium polyphosphate, and other fertilizers remained consistent.

**Table 3 plants-13-02282-t003:** RT-qPCR primer sequences.

Primer	R 5′ to 3′	F 5′ to 3′
ACT	TGCTTGACTCTGGTGATGGT	AGCAAGATCCAAACGGAGAATG
HK	GCGACGTTATGGTTGTAT	ACTCTTCCAACTCCCTCAACAC
IDH	GCAACCATCACACCAGATGAA	ACTCTTCCAACTCCCTCAACAC
G6PDH	ATGGATCTCCCTACCAGTCG	CTCATCAAAATAGCCACCTCT

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
