# Peer review of "Influence of Water and Fertilizer Reduction on Respiratory Metabolism in Sugar Beet Taproot (Beta vulgaris L.)"

_plants, 2024, doi:10.3390/plants13162282_

Round 1

Reviewer 1 Report

Comments and Suggestions for Authors

109 line should be corrected: ..2023.Soil..

Author Response

Comments 1:  109 line should be corrected: ..2023.Soil..

Response 1: Thank you for pointing this out. I agree with this comment. Therefore, I have made changes to the issue.[109 line]

[updated text in the manuscript if necessary]

Reviewer 2 Report

Comments and Suggestions for Authors

Line 118: Please to give name of the fertilizer, its composition, and amount of the supply.

Line 118: For water irrigation, was is at field capacity, please to give details. This is important for reproductivity regarding water reduction %age.

For fertilizer and water % age of reduction please to give corresponding amounts between (. ).

Nitrogen and Phosphorus:  what is the amount  corresponding to 100%. Please to give amounts corresponding to %ages of the fertilizers added to crop.

Material and methods:

 Give name of each apparatus as “model (manufacturer, country)” used in the investigation: water meters., gas analyzer. etc

Line 152: please to specify parameters measured for Energy charge, please to homogenize for this name as we found EC in results section. What are units for each measured parameter?, please to make it clear especially for fig3D.

Line 153: please to give methods and references for determination of these parameters ATP, PK, etc .

Line 176: Results: for most of paragraphs in this section, please to compare by providing values or % of differences between treatments, significant increases or decreases in comparison with control, or values (%)ages for highest/lowest effects.. This was well done in 3.1.

Table 2: complete the table with data, if two or more treatments have the same amount of a treatment, please to make it clear, for example, what was N, P and K amounts in CK, was N of 135 for F3W2, F3W1 and F2W3, please to check for clarity. In the bottom of the table, please to add, meanings of the abbreviations.

Figure 3: Is there an unit for EC ? (please to see remark Line 152).

Figure 6: please to check for Y axis, give words in English , and give significance of abbreviations in the figure caption.

Comments on the Quality of English Language

Only minor editing of English language required

Reviewer 3 Report

Comments and Suggestions for Authors

Review – plants-3088630

The manuscript contains a number of interesting results. It expands our knowledge about the biochemical processes taking place in the (storage) roots of sugar beet. Unfortunately, it is not yet at a level fully acceptable for printing. In my opinion, the article should focus more on achievements at the biochemical level rather than referring to agricultural practice. In its current form, there are more questions and comments regarding agronomic issues. It is also unclear how the obtained results can help in agricultural practice. Authors should also clearly highlight new and original results.

 Specific comments and suggestions

11.       Titel - I suggest specifying in the title what part of the plant was examined

22.       Line 14-15, water and fertilization reduction relative to what? Recommended and/or practiced fertilization and irrigation?

33.       Line 15, please remove the sentence "The goal……..beet industry" Fertilizer doses are determined taking into account the nutritional needs of plants and the possibility of supplying plants with nutrients from the soil.

44.       Line 11 and 17, There was no need to write "inner Mongolia" twice

55.       Line 21 and 29, the abstract also repeats the obtained result unnecessarily, moderate reduction in water and fertilization do not negatively....

66.       The sentences in lines 21 and 22 are contradictory, either a moderate reduction in irrigation and fertilization had a significant effect or not!

77.       Line 24, only the enzyme abbreviations are given, without the full name

88.       Line 15, it should be specified whether the research concerned the entire root or only its part defined as storage roots or taproots. This comment applies to the methodology and the entire article. The research material should be precisely defined.

99.       Line 31-33, it is difficult to agree with the content of this sentence. How can the results be translated into practical recommendations? It seems better to highlight what is new in the research.

110.   Line 78, 80, abbreviations for organic compounds require development, at least the first time in article. Please consider making a list of abbreviations and their full names.

111.   Line 89-100, lacks a clearly formulated scientific hypothesis, followed by a specification of objectives to verify the hypothesis. Currently, the paragraph is too general and the significance of the results for practice is controversial. How will they help the farmer determine the optimal nitrogen dose?

112.   Line 102-110 and Table 1, full soil characteristics are missing (soil texture and soil taxonomic unit); methods should be listed under the table. How was available NPK determined? Moreover, there is no assessment of the NPK content in the soil. The table shows that there was a lot of N available in the soil. This would explain why the reduced N dose was better than CK.

113.   Line 114-128, no reference in the text to table 2; whether the same fertilizers were used at each date (phase).

114.   Line 14, Test Materials – remove please; sentence 114-116 can be placed on line 128.

115.   Line 118 – treatment CK – control group - How were irrigation and fertilization levels determined? Did this actually reflect the plants' needs?

116.   Furthermore, please consider replacing the abbreviation CK with the abbreviation F3W3 (Table 2). In my opinion, it is a logical consequence of the way in which treatments are described.

117.   Table 2- please write the doses in pure P and K, not in the form of oxides

118.   Unit kg hm-2 ? The units should be given according to the SI system

119.   Line 139-146, Please additionally specify the stage of plant growth and development according to the phenological key;  specify what part of the root was examined.

220.   Are 3 plants (specifically roots) enough to make a reliable assessment? How was plant material prepared for chemical analysis after thawing?

221.   Line 149 - save the algorithm according to the editorial requirements.

222.   According to the requirements of the Plants journal, the methodological part of the article comes after the Discussion chapter

223.   Line 155, 158, etc., always provide the manufacturer and country of the manufacturer

224.   Line 173, use a unified way of writing the significance level "p < 0.05". The capital letter P also appears in the text

225.   Why was one-way ANOVA chosen? Two-way ANOVA can also be used for a split-plot system. It would therefore be possible to determine whether there was an interaction of factors, and which factors had a greater impact.

226.   Subchapter titles can be shortened, e.g. Line 233 - Energy metabolism in sugar beet taproots

227.   Figure 3, please write in the title of the figure whether the results are averages for two years?

228.   Discussion - the purpose of growing sugar beet is to produce sugar. Therefore, all over the world, efforts are being made to improve the quality of roots in terms of sucrose content (among other things). The authors do not mention anything about this fact in the discussion. The increased rate of respiration causes the loss of this compound in the taproots. Moreover, the rate of respiration in the roots, energy charge, etc. directly depend on photosynthesis. The studies conducted do not provide any information about leaf mass and its development. Excess nitrogen causes intensive leaf growth, often negatively affecting the weight of taproots and quality.

229.   Line 482-489, subsection "Conclusion" is more of a summary than conclusions. It's more like an abstract. The conclusions need to be improved. The metabolism in beet roots changes during the growing season, so which variant is optimal for yield and high quality? How can the acquired knowledge be used in agricultural practice?

Round 2

Reviewer 3 Report

Comments and Suggestions for Authors

The manuscript has been revised. Most of my comments and remarks have been taken into account. Some of the answers are debatable, but I accept them (as well as manuscript). This is especially true for statistical analysis. The split-plot experimental design suggests that the authors wanted to find out the effect of the interaction of factors. Two-way ANOVA also allows us to determine the effect of a single factor.

In the current manuscript, the following should still be improved:

1. the methodological part concerns soil properties. Determination of the total form of P and K in soil requires a specific chemical method (e.g. combustion, digestion in what reagent?). In the current form, we only learn that the analytical method was used (e.g. flame photometry). For available phosphorus, the Olsen method is specified, and for potassium?

2. Regarding the presented algorithms,

The formula for "Respiration rate" should be on a separate line ending with the equation number in brackets (please check MDPI publications)
